# The Current State of Radiotherapy for Pediatric Brain Tumors: An Overview of Post-Radiotherapy Neurocognitive Decline and Outcomes

**DOI:** 10.3390/jpm12071050

**Published:** 2022-06-27

**Authors:** Nicholas Major, Neal A. Patel, Josiah Bennett, Ena Novakovic, Dana Poloni, Mickey Abraham, Nolan J. Brown, Julian L. Gendreau, Ronald Sahyouni, Joshua Loya

**Affiliations:** 1School of Medicine, Mercer University School of Medicine, Savannah, GA 31404, USA; nicholas.allen.major@live.mercer.edu (N.M.); neal.atul.patel@live.mercer.edu (N.A.P.); josiah.bennett@live.mercer.edu (J.B.); enovakovic3@gmail.com (E.N.); 2Department of Surgery, Eisenhower Army Medical Center, Augusta, GA 30905, USA; dana.m.poloni.mil@mail.mil; 3Department of Neurosurgery, University of California San Diego, San Diego, CA 92103, USA; mickey.abra@gmail.com (M.A.); ronald.sahyouni@gmail.com (R.S.); jjloya@health.ucsd.edu (J.L.); 4Department of Neurosurgery, University of California Irvine, Orange, CA 92868, USA; nolanb@hs.uci.edu; 5Johns Hopkins Whiting School of Engineering, Baltimore, MD 21218, USA

**Keywords:** clinical outcomes, glioblastoma, glioma, medulloblastoma, neurocognitive decline, pediatric, pilocytic astrocytoma, quality of life, radiotherapy, survival, tumors

## Abstract

Tumors of the central nervous system are the most common solid malignancies diagnosed in children. While common, they are also found to have some of the lowest survival rates of all malignancies. Treatment of childhood brain tumors often consists of operative gross total resection with adjuvant chemotherapy or radiotherapy. The current body of literature is largely inconclusive regarding the overall benefit of adjuvant chemo- or radiotherapy. However, it is known that both are associated with conditions that lower the quality of life in children who undergo those treatments. Chemotherapy is often associated with nausea, emesis, significant fatigue, immunosuppression, and alopecia. While radiotherapy can be effective for achieving local control, it is associated with late effects such as endocrine dysfunction, secondary malignancy, and neurocognitive decline. Advancements in radiotherapy grant both an increase in lifetime survival and an increased lifetime for survivors to contend with these late effects. In this review, the authors examined all the published literature, analyzing the results of clinical trials, case series, and technical notes on patients undergoing radiotherapy for the treatment of tumors of the central nervous system with a focus on neurocognitive decline and survival outcomes.

## 1. Introduction

Central nervous system tumors are the leading cause of cancer mortality in childhood and are the second-most-common malignancy diagnosed in this patient population behind leukemia [1,2]. Intracranial tumors contribute to 20% of all pediatric malignancies with an incidence of 3.7 cases per 100,000 children [3,4,5]. Initial maximal safe resection is the standard of care for most of intracranial tumors, with prognosis dependent upon extent of resection for medulloblastoma [6], craniopharyngioma [7], ependymoma [8], low-grade glioma [9], and high-grade glioblastoma [10]. However, invasion into eloquent anatomy can prohibit maximal resection and, consequently, maximal progression-free survival [9]. Multimodal-treatment strategies potentially encompass early surgical resection, local radiation, complete craniospinal radiation, and chemotherapeutic treatments to achieve the maximal clinical outcome. Local control potentially improves with adjuvant radiotherapy for many pediatric brain tumors and is used in the management of a substantial fraction of solid malignancies [11].

While advances in multimodal therapy have improved outcomes [12], they may come at the cost of long-term, late effects [13], such as neurocognitive deficits [14,15], hearing loss [16], endocrine dysfunction [17], secondary malignancy [18,19], and radiation necrosis [19,20]. Neurocognitive dysfunction was recognized early on as a late effect of craniospinal irradiation when, in 1969, HJG Bloom noted a high rate of dementia in a cohort of 82 patients with medulloblastoma, who received cranial or craniospinal radiotherapy before the age of 2 [21]. In totality, late effects contribute to reduced direct and indirect health-related quality of life (HRQoL) [22,23,24,25,26,27], with neurocognitive dysfunction identified as the leading cause of reduced quality of life in long-term survivors of pediatric brain tumors [28].

There is well-established literature on the developmental consequences of these late effects, with children at increased risk for cognitive, emotional, and behavioral decline [14,29,30], regardless of focal [31,32,33] or whole-brain irradiation [29,30,34,35]. When compared to surgery or chemotherapy alone, radiotherapy is consistently associated with poorer neuropsychological outcomes [29,36]. Indeed, a cross-sectional survey of 342 patients with brain tumors treated with radiotherapy and 479 sibling controls showed the former patients were 28.8 times less likely to drive a car and 10.8 times less likely to be employed compared to their siblings [37]. In addition, these patients are found to have increased chances of being involved with theft, fraud, or assault [38,39]. The severity of these morbidities is directly related to dosage of radiation and volume of normal tissue exposed to radiation, while inversely related to age at time of irradiation [17,37,40,41,42,43,44,45]. Tumor size, location, baseline intellectual function, and enzyme polymorphisms with genetic predisposition also influence elements of cognitive function [46,47].

To help mediate against these long-term sequelae, approaches for prevention of and reduction in neurocognitive dysfunction include avoiding the use of radiotherapy in young children and reducing the dose and volume of brain that is irradiated [28]. Development of intensity-modulated radiotherapy (IMRT), volumetric-arc therapy (VMAT), and proton-beam radiotherapy (PBRT) over the past decades have sought to maximize the dose coverage of the tumor, while minimizing coverage to nearby normal tissue when compared to traditional techniques, such as conformal radiotherapy [48].

Improvements in long-term survival further underscore the impact of these late effects, as survivors now increasingly have longer lifespans, wherein, they must contend with these morbidities [48]. With the concomitant rise in prevalence, opportunities emerge to further understand the nature and consequences of the late effects of radiotherapy deeper into the patients’ course of disease. In this review, the authors examined all the published literature analyzing results of clinical trials, case series, and technical notes on patients undergoing radiotherapy for the treatment of tumors of the central nervous system, with an emphasis on neurocognitive decline and survival outcomes.

## 2. Survival Outcomes and Benefit of Radiotherapy

### 2.1. Medulloblastoma

With approximately 500 children being diagnosed each year in the United States, medulloblastoma is the most common malignant brain tumor of childhood and is the most common embryonal tumor [49,50,51]. As with other tumors, medulloblastoma usually presents with signs and symptoms of increased intracranial pressure secondary to obstructive hydrocephalus [50]. Additionally, 6%–32% of presentations are metastatic [52,53]. Initial work-up includes magnetic resonance imaging (MRI) that will classically reveal a contrast-enhancing cerebellar-midline mass [54,55]. Radiotherapy is an essential element in the treatment of pediatric medulloblastoma [51]. In the past two decades, advances in radiotherapy have included the development of a better-target volume delineation using imaging and use of adaptive techniques. IMRT, VMAT, image-guided radiotherapy, and particle-beam therapy [56,57,58]. Current dose-fractionation protocol includes 23.4 Gy in 13 fractions of craniospinal irradiation with a posterior fossa boost of 30.6 Gy in 17 fractions (Table 1) [59,60,61,62]. However, a new clinical trial has noted that current protocols lead to significant post-radiotherapy neurotoxicity. Thus, a reduction in dose or in some cases, elimination of craniospinal irradiation may be necessary. The clinical trial has proposed a new protocol for WNT 2-driven medulloblastoma-radiotherapy dosing: 18 Gy in 10 fractions plus a tumor-bed boost of 36 Gy in 20 fractions for a total primary site dose of 54 Gy in 30 fractions without concurrent chemotherapy followed by the standard six cycles of adjuvant systemic chemotherapy. By reducing the dosage to the craniospinal axis and reducing the total tumor-bed dose, the investigators are hoping this new protocol will decrease the late side effects of craniospinal irradiation and morbidity [63].

Prior treatment protocols emphasized higher radiation dosages to achieve higher cure rates, however contemporary protocols utilize risk-based dosing after accounting for the late effects of neurotoxicity; higher-risk patients will still receive high dosages [74], while standard risk indicates a lower dose [14,40,75,76,77,78]. Current treatment for medulloblastoma may involve gross total resection followed by radiotherapy and platinum-based chemotherapy [78,79,80,81]. These patients are classified postoperatively as standard risk or high risk (>1.5 cm^2^ residual disease after resection, known metastatic disease, or anaplastic large-cell histology) [51,82]. Children under 3–5 years are treated with chemotherapy only [83,84,85]. Prospective studies have shown 5-year event-free survival at 80–82% for average risk [59] and up to 70% for high-risk patients with current management protocol [86,87].

Risk stratification has classically involved evaluating the extent of the disease and the age at diagnosis, with younger age conferring a negative prognosis [52]. A study of 188 children with medulloblastoma found that those with no evidence of metastatic disease had a significantly higher 5-year progression-free survival (PFS) at 70%, compared to those with microscopic tumor cells in the CSF (57%) or those with gross nodular seeding (40%) [88]. This same study found the PFS of children between 18 months and 3 years to be 32% compared to 58% for those older than three. Decreased amounts of radiation therapy used in children younger than three was thought to play a role in the difference between the two groups.

The 2021 World Health Organization (WHO) Classification of Tumors of the Central Nervous System has separated medulloblastomas into two main types: molecularly defined and histologically defined [89]. The first main classification, molecularly defined medulloblastomas, are divided into four subtypes based on respective markers: WNT-activated, SHH-activated and TP53-wildtype, SHH-activated and TP53-mutant, and non-WNT/non-SHH. The second main classification, histologically defined medulloblastoma, is under its own subtype as “histologically defined” [89]. A 2010 meta-analysis with 270 children younger than five years assessed the prognostic role of clinical parameters and histology of early-childhood medulloblastoma and found that the 8-year event-free survival (EFS) and overall survival (OS) were both significantly higher in those with desmoplastic/nodular and desmoplastic with extensive nodularity, compared to those with either classic or anaplastic medulloblastoma [90]. The four molecular subtypes of medulloblastoma may be divided by genetic mutation for the purposes of risk stratification. The Wingless-related integration site (Wnt) pathway-activation group has the best prognosis, with an overall survival rate of 95%, and is mainly found in WNT-activated medulloblastoma. However, the Wnt-pathway activation is only present in ~10% of cases [91]. The activation pathway for sonic hedgehog (SHH), present in 30% of cases, is found chiefly in infants and adults. The overall survival for those with this mutation is ~75% [91]. The worst prognosis belongs to the non-WNT/non-SHH pathways, where some tumors display a high-level overactivation of the MYC protooncogene, yielding overall survival of ~50% at five years [92].

Treatment of medulloblastoma typically involves a combination of surgery, radiation, and multiagent chemotherapy [50,93,94,95,96]. The surgeon can achieve a margin-free excision or a near-total excision in most cases. No investigators have completed prospective, randomized trials evaluating survival in those with total versus near-total excision to date. A 2018 systematic review of retrospective studies, assessing the impact of the extent of resection on survival, found 16 articles with 1489 patients that did show a statistically significant association and 20 articles with 2335 patients that showed no significant association between the two groups. The authors concluded that the prognostic importance of the extent of resection for medulloblastoma is unclear [92]. Another retrospective study of 787 patients with medulloblastoma separated patients by their molecular subgroup, and the predictive value of the increased extent of resection decreased when the different subgroups were considered [97]. Thus, the standard of care for maximal, safe surgical resection remains. Surgical excision of small, residual portions of medulloblastoma is not recommended, when the likelihood of neurological morbidity is high. There is insufficient evidence that total resection imparts a survival benefit compared to subtotal resection [97].

Radiation therapy is started following surgical excision and is limited by its toxicity [94,95,96,98]. Proton-beam irradiation is a newer technique that is equally effective compared to conventional radiation therapy in treating posterior fossa and craniospinal disease, while significantly reducing the irradiation dose to the surrounding structures [98,99,100]. In a study comparing photon and PBRT in children with standard-risk medulloblastoma, Eaton et al. observed no significant difference between 6-year recurrence-free survival and overall survival [62]. In addition, patients at average risk are given a radiation boost to the primary site because 50–70% of recurrences develop in the posterior fossa, usually in the tumor bed and surrounding leptomeninges [101,102].

A Children’s Oncology Group study was a phase III trial of 379 children who have undergone total or near-total resection of a medulloblastoma, without evidence of disseminated disease [59,103]. Following resection, patients were treated with 23.4 Gy of craniospinal radiation, followed by a boost to the posterior fossa of 32.4 Gy. Weekly vincristine was given during radiation therapy, and a combination of vincristine, cisplatin, and either lomustine or cyclophosphamide was given for eight cycles. No significant difference in efficacy was found between the two chemotherapy regimens [59,103]. Infections were slightly higher in the cyclophosphamide group, while electrolyte disturbances were more common in the lomustine group. Event-free survival was higher in both groups receiving adjuvant radiation and chemotherapy than past surgery and radiation trials alone [59,103]. All patients undergoing this treatment developed hematologic toxicity. Approximately 25% developed ototoxicity. A cumulative cisplatin dose has not shown an increased event-free or overall survival, so lower doses are recommended in the treatment of medulloblastoma [104].

High-risk patients with metastatic or unresectable disease are usually treated with more aggressive chemotherapy and radiation, although there is no consensus about the most optimal therapy. Another Children’s Oncology Group study of 161 children ≥3 years of age with high-risk medulloblastoma were treated with postoperative radiation along with carboplatin and vincristine, followed by six maintenance cycles of cyclophosphamide, vincristine, and cisplatin [105]. Maintenance chemotherapy with cisplatin had a five-year progression-free and overall survival rates of 59% and 68%, respectively [105]. There were no treatment-related deaths reported. Another new strategy in the treatment of metastatic disease is hyperfractionated accelerated radiotherapy, which can be used in combination with high-intensity chemotherapy [106].

Therapies targeting molecular markers of medulloblastoma are currently under investigation. Vismodegib is an antagonist of the smoothened receptor that causes a disruption of transcription factors in the sonic-hedgehog pathway. It is currently only FDA-approved for the treatment of basal-cell carcinoma. Clinical trials are currently underway, and the phase 1 trials have shown that vismodegib is well-tolerated but is associated with teeth and bone abnormalities, including osteonecrosis in those taking concurrent steroids [107]. A phase II trial of 12 patients with sonic-hedgehog mutations had only 4 show a response to vismodegib, so further studies are needed to identify molecular patterns most susceptible to smoothened receptor antagonism [108].

### 2.2. Low-Grade Gliomas

Pilocytic astrocytoma (PA) is the most common primary brain tumor of childhood and is typically found in the cerebellum, but it may also present in the third ventricle, spinal cord, optic pathways, and cerebral hemispheres [109]. Maximal surgical resection is the standard of care; however, multimodal therapy is indicated with unresectable tumors in central locations, such as the optic pathway [110], and with recurrent tumors [111]. The prognosis for PA is among the best for primary brain tumors with a 5-year survival rate of 94% [112]. Grouped initially due to their similar gross and radiographic features, pilomyxoid astrocytoma (PMA) was distinguished from PA in 1999 [113]. They are now separated based on their histological appearance. PA has a biphasic architecture with protoplasmic cells, Rosenthal fibers, and eosinophilic granular bodies that are rare or absent in PMA. PMA has a monophasic architecture with a myxoid background and an angiocentric pattern [114]. PMA is critical to distinguish from PA as it has a more unpredictable course and is potentially more aggressive [113]. Maximal surgical resection is the mainstay of therapy for both PA and PMA, and radiation therapy is reserved for those who begin to progress after initial tumor resection [64]. The role of chemotherapy in low-grade astrocytoma is still uncertain. Still, it is generally limited to those with recurrent disease and young children in whom the risks of radiation therapy outweigh those of chemotherapy [64].

### 2.3. High-Grade Gliomas

High-grade gliomas (HGG) are divided into anaplastic glioma and glioblastoma. Both are aggressive, malignant tumors with poor prognoses. Length of survival in those with glioblastoma is dependent on tumor location and extent of resection [115]. A small retrospective study of 27 pediatric patients with glioblastoma found that those with superficial tumors amenable to complete resection had a median overall survival of 106 months compared to 11 months for those with incomplete resection [115]. As with other neoplasia of the central nervous system, maximal surgical resection is preferred at the outset to confirm the diagnosis, grade the tumor, and begin molecular analysis. When the tumor is not amenable to surgery, stereotactic biopsy is preferred. Once maximal resection is complete, the current standard of care in the treatment of glioblastoma is radiotherapy with concomitant temozolomide, an alkylating agent, followed by adjuvant temozolomide [65].

Radiotherapy is the most effective nonsurgical treatment for gliomas, however, glioma recurrence is common as glioma cells are highly radioresistant. It is thought that cancer stem cells contribute to radioresistance in gliomas through the Notch signaling pathway. A study by Wang et al. is discussed below in detail and provides insight on how inhibiting the Notch pathway leads to more radiosensitive glioma cells [116]. Furthermore, it has been shown that glioma stem cells are more radioresistant than the non-stem glioma cells [117]. Although temozolomide may increase radiotherapy sensitivity of tumor cells, glioblastomas remain highly resistant to radiation [118]. Due to the highly tumorigenic activity of these stem cells, enhancing their radiosensitivity has been a target for improving overall effectiveness of radiotherapy. Chemotherapy with molecular targeting has shown promise in enhancing radiosensitivity. Bao et al. has shown that inhibitors of two DNA-damage-checkpoint kinases, Chk1 and Chk2, increased the cell death of glioma stem cells in response to radiation [119]. However, this target is not exclusively expressed by tumor cells, limiting the therapeutic index of inhibitor therapy by increased radiosensitivity of normal cells as well [120]. In more detail, the Notch signaling pathway has emerged as a target of radioresistance in cancer stem cells. Normal notch signaling promotes self-renewal and dedifferentiation in many adult stem cells in breast [121,122], intestine [123,124], and neuronal tissue [125,126,127]. Dysfunctional Notch activity is seen in many human tumors, such as breast cancer [128,129], leukemia [130], and glioma [131]. The Notch pathway is dependent upon a final proteolytic step, wherein a gamma-secretase releases the intracellular domain of the Notch protein (NICD) [132,133]. Notch activation via NICD1 overexpression promotes tumorgenicity [120], whereas inhibition by gamma-secretase inhibitors (GSI’s), which are used to block Notch activity in vivo and in vitro, reduces tumorigenicity [134]. In a study that measured the tumor size of CD133+ gliomas both in vitro and in vivo in response to radiotherapy with Notch inhibition via GSI’s or knockdown expression, Wang et al. found significant increases in cell death at clinically relevant doses. The degree of cell death in non-stem cells was unaltered by Notch inhibition as compared to the CD133+ stem cells when irradiated. Additionally, constitutive expression of the active intracellular domains of Notch1 or Notch2 protects glioma stem cells against radiation. Reduction in Notch1 or Notch2 levels also leads to radio-sensitive glioma stem cells. Thus, overexpression of Notch in cancerous glioma cells may play a critical role in the radioresistance of glioma stem cells [116]. Lower doses of radiation are required when tumor cells are molecularly radiosensitive, thus, further research into molecular targeted therapies can ultimately contribute to reductions in the late effects of neurotoxicity.

While limited randomized trials on the treatment of HGG exist, a 2020 case report of a 3-year-old girl diagnosed with glioblastoma highlights the importance of molecular targeted therapy in the future of care [135]. She initially presented with progressive seizure activity. Neuroimaging revealed a significant, heterogeneously enhancing, mixed cystic and solid mass in the left frontal-parietal-temporal region. She underwent debulking surgery, and genomic analysis of the tumor showed a BRCA2 nonsense mutation. Standard adjuvant radiation and temozolomide were used initially. Nine months after surgery, treatment with a combination of olaparib, an inhibitor of poly ADP ribose polymerase, and temozolomide was used for 16 cycles. She remains neurologically intact and “continues to experience an exceptional and durable response (>2 years)” after her treatment, with no evidence of tumor recurrence in serial imaging [135]. This case underscores the importance of genomic sequencing of brain tumors and the role of progressing to more targeted molecular therapies instead of systemic chemotherapy in the future.

### 2.4. Brainstem Gliomas

Brainstem gliomas are a subgroup of astrocytoma that range from low-grade lesions that may not require any intervention to high-grade malignancies with poor prognoses [136,137,138,139]. Diffuse Midline Glioma (DMG) is important, since radiation therapy is the mainstay treatment. A subtype of particular interest, diffuse intrinsic pontine glioma (DIPG) is a rare, aggressive type that forms in the brainstem of pediatric patients. With most survival rates at less than one year, immediate treatment is imperative. The first-line treatment for DIPG is radiation therapy, as surgical resection is not a viable option [140]. The molecular etiology of DIPG may be attributed to the “H3 K27M-mutant” in ~80% of cases, as noted in WHO 2016. However, WHO 2021 has adapted a new name, “H3 K27M-altered”, to recognize other mechanisms where the pathogenic pathway may be altered in these tumors [89,141]. Approximately 500 new cases of brainstem glioma are diagnosed in the pediatric population each year [142,143]. Neurofibromatosis type I is one of the only known risk factors for developing a brainstem glioma [144,145,146]. Patients may present with cranial-nerve deficits corresponding to the location of the mass [138,139]. Exophytic tumors of the dorsal brainstem may cause obstructive hydrocephalus [147]. The site of these tumors provides a unique challenge for their treatment [40]. Maximal safe resection remains the standard of care when possible [9,66]. Radiation therapy may be used in those patients whose tumor burden is not amenable to surgery or those where gross total resection is not achieved [67,68,69]. Radiation therapy for brainstem gliomas has been used sparingly due to its side effects. One study has shown that children who received either conventionally fractionated or hyperfractionated radiation for brainstem gliomas had significantly lower IQs than those who received surgery alone [148]. However, this is likely confounded by those with more severe tumor burden being the ones who received radiation. Other side effects of radiation to the brainstem include mild nausea and vomiting, endocrinopathies from damage to the hypothalamic-pituitary axis, and brainstem necrosis, which is a rare but devastating outcome [149,150,151]. Over the past 30 years, multiple trials have shown that chemotherapy has activity against brainstem gliomas and could be a safe first-line therapy for those in which the consequences of brainstem radiation is desired to be avoided [152,153,154,155]. Combinations of vincristine and platinum-based chemotherapeutics have been studied, but no consensus on the optimal regimen has been reached [152,153,154].

### 2.5. Ependymoma

Ependymomas are tumors of the lining of the ventricular system that typically arise in the fourth ventricle in the first two decades of life [109]. Intracranial ependymoma contributes to 10% of all intracranial tumor presentations in children [112]. Five-year survival approaches 50–80% [70,71,72,156,157]. The 2021 World Health Organization Classification separates ependymomas according to a combination of histopathology, molecular markers, and anatomic sites. The molecular groups are now divided across the following anatomic sites: supratentorial, posterior fossa, and spinal compartments. WHO 2021 subcategorizes molecularly defined types of supratentorial ependymoma into two types: ZFTA fusion and YAP1 fusion. There are also two major molecular subtypes of posterior fossa ependymomas: group PFA and group PFB. The spinal tumor-molecular subtype is characterized by the presence of MYCN amplification. Lastly, the histological ependymoma subtypes include Myxopapillary ependymoma and subependymoma [89].

As with other surgeries in the posterior fossa, complications of ependymoma resection include cerebellar ataxia, lower cranial nerve damage, and posterior fossa syndrome [158]. Following maximal safe resection, further treatment is guided by the extent of resection and tumor grade. Local or craniospinal radiation with doses of 54–59.4 Gy is usually given to those with subtotal resection [70,71,72]. Extent of resection is the most determinant factor of prognosis [8,156,159,160,161,162,163,164,165]. Should recurrence occur, it is most commonly local [8,159,161,163]. A long-term prospective study evaluated the health-related quality of life (HRQoL) of 40 patients < 4 who received proton-radiation therapy for a central-nervous-system tumor [166]. Ependymoma was the most common tumor in the cohort (*n* = 22), followed by medulloblastoma (*n* = 9). The median age at radiotherapy was 2.5, and the median age at follow-up was 9.1. The authors found that the HRQoL was variable with just over a third of patients and families achieving levels equivalent to healthy children. In total, 90% of the patients were able to function in a regular classroom. Patients who developed hydrocephalus or those that required feeding-tube placement reported significantly lower HRQoL. Scores among the cohort remained stable from the baseline analysis until the last follow-up [166]. A similar retrospective study examining patients with posterior fossa brain tumors found that families with lower socioeconomic status also reported more inadequate measures for their quality of life [167].

### 2.6. Craniopharyngioma

Craniopharyngiomas are supratentorial tumors of childhood that arise from the remnants of Rathke’s pouch and may contain a cystic component [168]. Like pituitary adenomas, craniopharyngiomas may present with headaches and bitemporal hemianopia [169]. Disruption of the hypothalamic–pituitary axis may lead to diabetes insipidus or another endocrine dysfunction [169]. These symptoms may be present for over a year before a diagnosis is made, as this tumor is slow-growing [169]. Historically, the management of craniopharyngiomas has been controversial, with some advocating for more aggressive surgical management to achieve the maximal extent of resection and others advocating for a more conservative approach in the operating room, followed by radiation therapy [73]. Even when confirmed by imaging, gross total resection can result in recurrence up to 20–27% [7,170]. When treated with a conservative surgical resection with adjuvant radiotherapy, 10-year progression-free survival has been reported up to 84–100% [171,172,173]. While there is no absolute consensus today, most agree that optimal management includes removing the greatest tumor burden possible without introducing any iatrogenic deficits [174]. This is highlighted by the approximately 50% of survivors that have long-term sequelae that negatively impact their quality of life [175].

## 3. Neurocognitive Late Effects

### 3.1. Decline in Neurocognitive Development

Intelligence quotient (IQ) has been used as a metric to measure changes in neurocognitive development after the treatment of pediatric brain tumors with a mean score of 100 and a standard deviation of 15 to 16 [40]. Declines in IQ are likely attributed to failure to develop neurocognitively at a rate expected for the patients’ age, rather than a loss of previously established development. Rate of IQ decline is correlated with younger age at time of treatment, volume of brain irradiated, dose of irradiation, female sex, and hydrocephalus [29,35,40]. Merchant et al. showed that radiation dosimetry can be used to predict patient IQ after conformal radiotherapy treatment of localized ependymoma [176]. While the effects of radiotherapy are mainly implicated in explaining neurocognitive decline, additional factors related to diagnosis and treatment course, such as posterior fossa syndrome and hydrocephalus, are also known to negatively influence neurocognitive development [177]. Specifically, following surgical resection of medulloblastoma, up to 29% of patients experience posterior fossa syndrome, or cerebellar mutism syndrome, which is characterized by diminished or absent speech, ataxia, hypotonia, and emotional lability. These symptoms may be exacerbated by radiotherapy [178]. Declines are associated with similar declines in measures of basic academic achievement [14,179,180], which may present as differences in performance, with standardized testing as early as one-year post-treatment. Declines may continue as late as ten years post-treatment [29,181]. Certain deficits due to neurocognitive decline may be delayed in presentation, due to normal ability being expected later during childhood development, such as speech (Table 2) [182].

### 3.2. Mechanism of Neurocognitive Decline

While the pathophysiology of radiation induced long-term CNS damage is not fully understood, hypotheses generally fall under the primary effects or secondary effects of radiation [40]. Primary effects may be caused by inciting damage to progenitor cells, vasculature, inflammatory cells, and stromal cells [187]. Another mechanism consists of the radiation-induced destruction of oligodendrocytes, which causes inadequate myelination and white-matter necrosis. Secondary effects include radiation-induced oxidative stress in the myelin membrane [184]. This is extremely pertinent in pediatric tumors, as full myelination of the cerebral cortex is achieved in early adulthood [188]. This mechanism has been further substantiated over the past few decades by new technologies that quantify the volumes of brain tissue [189]. For example, in a study that measured post-operative IQ after treatment of medulloblastoma with age-matched controls, the survivors of medulloblastoma had statistically significant lower volumes of white matter and IQ scores. IQ scores had a statistically significant inverse association with volumes of white matter [179]. Additional longitudinal studies show regions receiving an increased dose of radiation are associated with greater rates of decline in white-matter volume [185,186]. This mechanism could have a direct effect on processing speed, memory, and, in severe cases, cause ataxia, urinary incontinence, and dementia [183].

Certain eloquent areas are also associated with memory creation and cognition. In the subventricular zone of the lateral ventricles and the subgranular zone of the hippocampal gyrus, neurons and glial cells are produced that contribute to memory production. These areas form part of the limbic system [190,191]. Additional structures of the limbic system include the hippocampus, parahippocampal gyrus, and amygdala, which is located in the temporal lobe. One study reports worsening neurocognitive dysfunction when targeting the left temporal lobe with chemotherapy in children <7 years of age [192]. In adults, an additional study shows neurological sequelae can be spared when avoiding the hippocampi during radiotherapy; in children this has yet to be studied [193]. Finally, cerebellar neurons form closed loops with neurons in the prefrontal cortex, temporal lobes, and limbic structures. Damage to these neurons can lead to posterior fossa syndrome in medulloblastoma and neurocognitive dysfunction and should be avoided if possible [178].

### 3.3. Management of Late Effects

There remains a shortage of published studies that investigate interventions for survivors of pediatric brain tumors suffering from neurocognitive late effects. Of the available studies, interventions fall under three categories: cognitive remediation, pharmacotherapy, and environmental modifications [40]. A case study involving the use of a memory notebook to remediate severe memory impairment in a survivor found that the patient’s academic achievement, classroom attendance, and completion of assignments increased slightly, despite a continued, significant memory impairment [194]. Stimulants, such as methylphenidate, have been shown to be effective medications for attention deficit hyperactivity disorder (ADHD), through improvements in cognitive function [195]. A randomized, double-blind trial that investigated the effects of methylphenidate on survivors of both pediatric leukemia and malignant brain tumors with impaired learning found that the group assigned methylphenidate showed significantly greater improvements in tests of vigilance, a proxy for attention, compared to the placebo group [196]. However, another study that exclusively tested survivors of brain tumors who received craniospinal irradiation 3–12 years earlier found no significant immediate, or delayed, benefit from methylphenidate therapy [197]. Environmental interventions in children with neurocognitive delay is often underestimated in its efficacy [198]. The family dynamic survivors are recovering in have been shown to play a role in neurocognitive recovery from traumatic brain injury, with more dysfunctional environments being associated with slower recovery when severity of brain injury and other medical factors are controlled for [199]. While the neurodevelopmental consequences of brain irradiation in pediatric patients are not fully elucidated, the interventions indicated for treatment of these effects seem to be just as, if not more, poorly established.

The risk factors for late effects of neurotoxicity with PBRT are similar to photon-based therapy: being of a younger age as well as the volume of the brain and spine that were irradiated [200,201,202].

## 4. Future Directions

While advancements in photon-based radiotherapy, such as IMRT and VMAT, reduce neurotoxicity to normal brain tissue, PBRT has emerged as a particularly promising modality for minimizing late toxicity due to reductions in the dose to vital organs (Table 3) [41,203,204,205].

Macdonald et al. found that initial control of pediatric ependymoma and sparing of normal tissue was favorable with PBRT compared to IMRT, a form of photon radiotherapy [206]. In addition, Mirabell et al. found a reduced incidence of secondary cancer resulting from radiotherapy with PBRT in rhabdomyosarcoma and medulloblastoma patients [207]. Indeed, Zhang et al. calculate a lifetime risk of 0.18 for secondary cancer attributable to PBRT, much lower than that of photon-based radiotherapy [209]. Eaton et al. found no significant difference in recurrence free survival or overall survival in standard-risk medulloblastoma patients treated with proton versus photon-based radiotherapy, however the need for endocrine replacement was significantly lower in those receiving PBRT [58]. A retrospective study of children with medulloblastoma treated with photon versus PBRT showed no difference in clinical outcome between groups, despite the cochlea receiving a lower mean dose in the PBRT group. It was hypothesized that the rates of ototoxicity did not differ between groups due to confounding ototoxicity from cisplatin therapy.

The initial biological effect of PBRT is considered to have the same theoretical relative biological effect (RBE) of 1.1 as photon-based therapy [210,211,212]. It is hypothesized that reduced incidence and/or intensity of late effects can be observed with PBRT due to the intrinsic physics of its dose deposition [41,213,214,215,216]. Proton beams exhibit a sharp peak in proton deposition called the Bragg peak, which is targeted to cover to tumor volume (Figure 1). The deposition before the peak is suppressed and the deposition distal to the peak is near zero, effectively minimizing the radiotherapy dose to normal tissue within the beam trajectory [41,217,218,219,220]. PBRT has been shown to reduce the dose to the brain, brainstem, optic nerve, and optic chiasm [221,222,223]. Even the small, critical organs that are typically susceptible to radiotherapy toxicity, such as the cochlea and hypothalamus, can be spared with PBRT, which preserves hearing, intelligence, and endocrine function [41,200].

Evidence of PBRT’s benefit towards reduced neurocognitive toxicity has grown over the past two decades [200,215,224,225,226,227,228,229,230,231,232]. Studies that measured neurocognitive outcomes post PBRT therapy did not find any significant neurocognitive impairment in survivors [200,201,202,208]. Processing speed was identified as the domain of greatest vulnerability [208,228]. A study by Kahelley et al. comparing cognitive outcomes after photon versus proton radiotherapy showed no significant decline in IQ among survivors receiving PBRT, while those treated with photon radiotherapy did show a significant decline [208]. The risk factors for late effects of neurotoxicity with PBRT are similar to photon-based therapy: being of a younger age as well as the volume of the brain and spine that were irradiated [200,201,202].

With the advent of these novel radiotherapeutic techniques coupled with developments in advanced MRI techniques, these novel approaches need to be tested in prospective studies of children with brain tumors. Primary outcomes should be survival as well as long-term neurocognitive function. Additionally, these studies should locate the tumor, gather its proximity to vital eloquent areas and tracks of the brain, and monitor outcomes as it pertains to location. With data from these prospective studies, optimal therapeutic strategies can be created for each specific type of brain tumor.

## 5. Conclusions

The management of pediatric central-nervous-system tumors with radiotherapy allows for greater lifetime survival but also lengthens the morbidity from late effects, especially neurocognitive decline. This is largely due to the unintended consequence of radiation-induced damage to the normal neural tissue caught in the beam trajectory. Advancements in photon-based radiotherapy confined the volume of the brain that was irradiated to fit the tumor margins more closely; however, the advent of PBRT further allows for greater sparing of normal neuronal tissue, by maximizing energy deposition at the tumor location while minimizing energy deposition both proximal and distal to the tumor target. While additional advancements in radiotherapy technology are certainly warranted for both an increase in survival and a decrease in neurocognitive late effects, further research is needed to develop interventions for those already suffering from these late effects. Current intervention options are limited and of uninspiring efficacy, mostly limited to environmental accommodations and cognitive remediations. Interventions for generalized developmental delay may attenuate the consequences of these late effects, but there remains an opportunity to implement specific strategies to intervene with survivors treated with radiotherapy, especially due to the emergence of consensus deficits in neurocognition, such as processing speed.

## Figures and Tables

**Figure 1 jpm-12-01050-f001:**
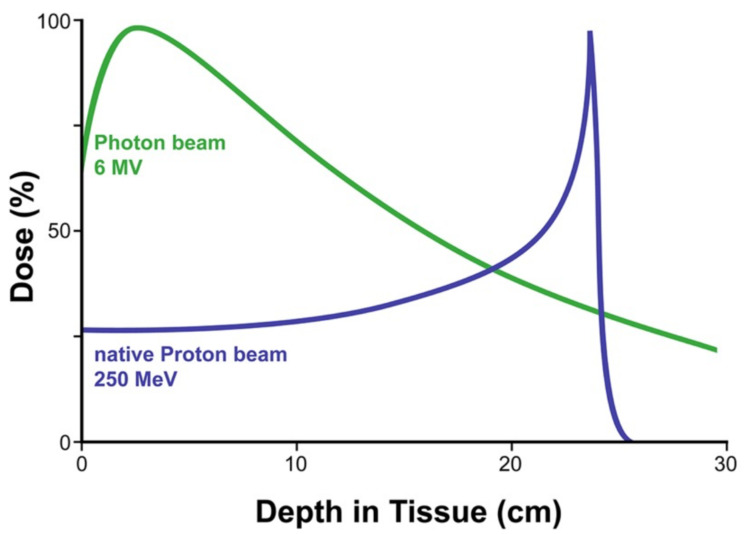
Graphical depiction of proton-beam dosage per tissue depth. The figure demonstrates that the dosage of the proton beam is higher in deeper tissues. The more superficial tissues are spared from increased dosages (blue line). In contrast, photon-beam radiation (green line) has increased dosage in superficial tissue with decreased dosage in deeper tissues.

**Table 1 jpm-12-01050-t001:** Description of pediatric-tumor treatments including proposed management and dosages.

Tumor Type	Mainstay Treatment	Specific Radiation Therapy	Reported Survival
Medulloblastoma	Maximal surgical resection with radiation and multiagent Chemotherapy [59,60,61,62].	Proton-beam-radiation therapy (23.4 Gy in 13 fractions of craniospinal irradiation with a posterior fossa boost of 30.6 Gy in 17 fractions)	86% at 5 years [59], 83% at 61.2 months [60], 78% at 4.8 years [61], 87.6% at 6 years [62]
Low-Grade Glioma (Pilocytic Astrocytoma)	Maximal surgical resection. Multimodal therapy for unresectable tumors and recurrent tumors [64].	Proton-beam radiation	>90% at 10 years [64]
High-Grade Glioma	Maximal surgical resection. Stereotactic biopsy if unresectable. Radiotherapy with concomitant temozolomide for non-surgical cases [65].	Proton-beam-radiation strength, dependent on radiosensitivity of tumor cells	26.5% at 2 years [65]
Brain Stem Glioma	Resection when possible and radiation therapy [9,66,67,68,69].	Proton-beam radiation	Widely variable: 34% at 5 years [68], 100% at 5 years [66], 33% at 5 years [69]
Ependymoma	Maximal surgical resection. Local or craniospinal radiation in those with subtotal resection [70,71,72].	Proton-beam radiation (54–59.4 Gy)	76% at 10 years [71], 74.8% at 5 years [72]
Craniopharyngioma	Controversial surgical resection with adjuvant radiotherapy [73].	Proton-beam radiation	88% at 5 years [73]

**Table 2 jpm-12-01050-t002:** Summary of the late effects due to radiotherapy on neurocognition.

Key Points of Discussion on the Neurocognitive Late Effects from Radiotherapy
Decline in IQ and academic achievement [29,35,36]. Decline in processing speed and memory [183]. Early-Onset dementia [183]. Destruction of white brain matter, oligodendrocytes, and other neurons [184,185,186].

**Table 3 jpm-12-01050-t003:** Comparison of complications (favorable or unfavorable) in the different radiotherapy modalities.

Type of Radiotherapy	Proton-Beam Radiotherapy	IMRT (Photon)	VMAT (Photon)
Comparison of Complications	Reduction in dose to vital organs [41,203,204,205]. Favorable sparing of normal brain tissue in pediatric ependymoma [206]. Reduced incidence of secondary cancer in medulloblastoma [207]. No significant decline in IQ [208]. No significant impairment in neurocognitive function in [200]. Reduction in risk of secondary cancer [209].	Reduction in neurotoxicity [41,203,204,205]. Unfavorable sparing of normal brain tissue in pediatric ependymoma [206]. No significant neurocognitive impairment in survivors [200,201,202,208]. Increased need for endocrine replacement [58]. Significant neurocognitive impairment [200,201,202,208]. 18% lifetime risk of secondary cancer [209].	Reduction in neurotoxicity [41,203,204,205]. Significant neurocognitive impairment [200,201,202,208]. Increased need for endocrine replacement [58].

Both IMRT and VMAT are subtypes of photon-based radiotherapy. IMRT = intensity modulated radiotherapy. VMAT = volumetric modulated arc therapy.

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
