# Peer review of "The Current State of Radiotherapy for Pediatric Brain Tumors: An Overview of Post-Radiotherapy Neurocognitive Decline and Outcomes"

_jpm, 2022, doi:10.3390/jpm12071050_

Round 1

Reviewer 1 Report

The manuscript entitled “The Current State of Radiotherapy for Pediatric Brain Tumors: An Overview and Update of Neurocognitive Decline and Survival Outcome” by Nicholas Major et al. carefully reviewed the current state of radiotherapy for pediatric tumors. The premise of the review topic is very interesting, however in its present version, the manuscript requires several significant areas of improvement before consideration for publication.

In my opinion major improvements are required

1) The list of radiation strategies for different types of brain tumors as a table should be added.

2) The quality of Figure 1 needs to be improved. It is not easy to read and understand.

3) Figure legends should be self-explanatory and properly abbreviated. More explanation is necessary.

4) Showing future directions as a figure will be a good idea.
5) I wonder if authors would provide a box (containing some bullet points) addressing some major points/ mechanisms/ challenges and/ or answers of some demanding questions of the discussed area.

6) Title “The Current State of Radiotherapy for Pediatric Brain Tumors: An Overview and Update of Neurocognitive Decline and Survival Outcome”, I think a more appropriate title should be found.

7) The conclusions of the review should be summarized as a model Figure.

8) Introduction needs improvement.

9) There are some points either discussed haphazardly or overlooked, that need to be discussed properly. For example, information about the radiation resistance is needed.
10) The quality of the figures is not good. It must be improved.
11) The text needs careful proofreading. There are many grammar and spelling mistakes.

Author Response

Editor 1:

  • The list of radiation strategies for different types of brain tumors as a table should be added.

- This is a great idea also echoed by the second editor. Thank you for the excellent suggestion. We added Table 1 as requested on lines 107-121 listing radiation strategies for each brain tumor type with citations.

2) The quality of Figure 1 needs to be improved. It is not easy to read and understand.

- Thank you kindly for this edit. We agree. The second editor agrees, as well. Unfortunately, due to resource limitations, we decided to remove figure 1 at this time, previously on lines 52-55.

3) Figure legends should be self-explanatory and properly abbreviated. More explanation is necessary.

- Removed figure 1, previously on lines 52-55. The previous “Figure 2” was updated to “Figure 1” with additional explanation within lines 584-589 as advised. Thank you for this correction.

4) Showing future directions as a figure will be a good idea.

- Sincere thank you for this idea to make a figure out of future directions. After further discussion with the team, we may opt to create a table with future directions as our resources may be limiting our author(s) to create a figure. However, we did add a concise table (Table 3 within lines 547-552) to demonstrate future directions of radiotherapy by comparing their respective complications.

5) I wonder if authors would provide a box (containing some bullet points) addressing some major points/ mechanisms/ challenges and/ or answers of some demanding questions of the discussed area.

Many thanks for this recommendation on a box that address major mechanisms of discussion. We added a box with bullet points to concisely address this in Table 2, within lines 464-465.

6) Title “The Current State of Radiotherapy for Pediatric Brain Tumors: An Overview and Update of Neurocognitive Decline and Survival Outcome”, I think a more appropriate title should be found.

- Thank you for this recommendation. The title has been changed to “The Current State of Radiotherapy for Pediatric Brain Tumors: A Review of Post-Radiotherapy Neurocognitive Decline and Survival Outcomes”

7) The conclusions of the review should be summarized as a model Figure.

- Thank you for pointing this out. Due to resource limitations, we were unable to create a model figure, however, we are hoping our newly created tables address this issue. Thanks again.

8) Introduction needs improvement.

- Thank you kindly for this correction. The introduction was proofread and corrected for grammar, spelling, and any other confusing wording.

9) There are some points either discussed haphazardly or overlooked, that need to be discussed properly. For example, information about the radiation resistance is needed.

- Many thanks for pointing out the deficient information on radiation resistance. I see where the issue described is and it seems the author(s) presented the information in a confusing way. A general introduction about radiation resistance has been added within lines 311-316, and hopefully a smoother, more organized transition into the discussion of the information has been added within lines 335-338.

10) The quality of the figures is not good. It must be improved.

- Thank you kindly for this pointing out this deficit. We authors agree. The second editor agrees, as well. Unfortunately, due to resource limitations, we decided to remove figure 1 at this time, previously on lines 52-55.

11) The text needs careful proofreading. There are many grammar and spelling mistakes.

- Many thanks for carefully reading the paper and identifying deficiencies in our grammar. We carefully proofread and edited for spelling and grammar throughout.

Reviewer 2 Report

Major and colleagues provide a review of "The Current State of Radiotherapy for Pediatric Brain Tumors: An Overview and Update of Neurocognitive Decline and Survival Outcomes." Indeed, the study is of interest to experts since it describes/discusses/highlights critical challenges in pediatric oncology and makes readers question the current standard care therapy for pediatric brain tumors. However, the review still needs to address and update relevant topics to the pediatric brain tumor field. Therefore, I would recommend modifying and revising the following:

Major

1) Please adapt all information derived from 2016 to the new WHO2021 (Louis et al., 2021),

2) Please update the names of medulloblastoma to WNT molecular subgroup/SHH molecular subgroup/ and on. It could also be, e.g., Wnt-activated MB (although less recommended). Can authors find literature and clinical trials that use the MB molecular subgroup information to adapt their therapy protocol? Are there any clinical trials (e.g., NCT04474964)? Please expand about the cerebellar mutism

3) What is out there for EPN molecular subgroups? Please adapt the new molecular classification (2021), e.g., ST-ZFTA

4) Are there any protocol adaptations for K27M, G34R/V, IDH mutant/wild type for gliomas? Please explore more about it.

Please improve the term "Malignant Gliomas." It is not recommended. 

5) Please also explore diffuse midline gliomas, including diffuse pontine gliomas.

6) Figure 1. Should be improved: Please make it readable with good resolution.

7) I strongly recommend making a table and categorizing for each tumor type:

What literature has about a tumor type/ Therapy tested/Final outcome

If the authors address all points above, the manuscript may reach the standard quality for publication in JPM/MDPI.

Author Response

Editor 2:

1) Please adapt all information derived from 2016 to the new WHO2021 (Louis et al., 2021),

 - Thank you very much for identifying this improvement with the updated citation. We adapted information from the 2016 WHO to the 2021 WHO and updated the citations throughout. Thanks again.

  • Please update the names of medulloblastoma to WNT molecular subgroup/SHH molecular subgroup/ and on. It could also be, e.g., Wnt-activated MB (although less recommended). Can authors find literature and clinical trials that use the MB molecular subgroup information to adapt their therapy protocol? Are there any clinical trials (e.g., NCT04474964)? Please expand about the cerebellar mutism

- Sincere appreciation for keeping the information up to date. We went through and made appropriate updates to molecular subgroups where WHO 2021 had updates. Within lines 478-481, we also added an expansion on cerebellar mutism. Lastly, we found the literature as suggested and discovered a potential update to the protocol. The trial suggested above is still recruiting, however, the plan and goal of the trial may greatly improve late side effects and outcome. We included the newly adapted protocol from this trial to the WNT-2-driven MB in lines 135-142. Thanks again.

  • What is out there for EPN molecular subgroups? Please adapt the new molecular classification (2021), e.g., ST-ZFTA

- Dear editor, thank you so much for identifying the new molecular classification of EPN subtypes. The WHO 2021 defines the molecular subtypes as recommended above (e.g., ZFTA) and this will greatly improve and update the paper. We have adapted the new molecular classification within lines 413-420 as suggested above. Sincere appreciation for pointing out this improvement.

4) Are there any protocol adaptations for K27M, G34R/V, IDH mutant/wild type for gliomas? Please explore more about it.

- Thank you very much for the suggestion to explore protocol adaptations for K27M, G34R/V, and IDH mutat/wild type gliomas. After exploration, Unfortunately, we did not find new protocol adaptations for these mutants, so we retained the original citations and author work.

Please improve the term "Malignant Gliomas." It is not recommended. 

- Thank you for this suggestion. We Improved to the term “Malignant Gliomas
“ to “High-Grade Gliomas” on line 298. Many Thanks.

5) Please also explore diffuse midline gliomas, including diffuse pontine gliomas.

- Sincere thank you for the suggestion to explore diffuse midline gliomas, especially diffuse pontine gliomas. We really enjoyed incorporating this information into the paper on lines 382-390 under the “Brain Stem Glioma Section 2.4” with appropriate citations. Thanks again.

6) Figure 1. Should be improved: Please make it readable with good resolution.

- Thank you kindly for this pointing out this deficit. We authors agree. The second editor agrees, as well. Unfortunately, due to resource limitations and time constraints, we decided to remove figure 1 at this time, previously on lines 52-55.

7) I strongly recommend making a table and categorizing for each tumor type:

What literature has about a tumor type/ Therapy tested/Final outcome

Dear Editor, thank you for this excellent recommendation to create a table for each tumor type. Both editors have suggested this and we authors immediately agree. We have added a table categorizing each tumor type with radiation therapy tested within lines 107-121. Many thanks.

Done.

Round 2

Reviewer 1 Report

The authors substantially improved the manuscript, which can now be accepted for publication

Author Response

Thank you for the kind words.

Reviewer 2 Report

The authors addressed all points besides the new table needs to be improved:

The term "Diffuse Midline Pontine Glioma" is unusual. Use Diffuse intrinsic pontine glioma (DIPG). For tumors occurring in the pons (DIPG) and in the midline structures (e.g thalamus) it is assigned as Diffuse midline gliomas (DMG). DIPG is part of the DMGs. The table does not provide Survival outcomes. Please Specify it.  If authors address these points, I have no other comments.

Author Response

Tables have been updated with respective survival numbers of the quoted studies. Tables finalized throughout. Also, the naming was fixed of "Diffuse Midline Pontine Glioma."